# Watch Your Step: Information Injection in Diffusion Models via Shadow Timestep Embedding

An Huang [1]   Junggab Son [1]   Zuobin Xiong [1]

## Abstract

Diffusion models have become the foundation of modern generative systems, with most research focusing primarily on improving generation efficiency and output quality. The timestep embedding component is a crucial part of the diffusion pipeline, which provides a temporal conditioning signal to the denoising network, enabling it to adapt its predictions across different noise levels throughout the process. Despite their potential to contain substantial information, timestep embeddings remain underexplored in current research, especially for security risks and reliable provenance. To fill this gap, we introduce **Shadow Timestep Embedding (STE)**, a novel mechanism that investigates the underutilized temporal space for malicious information injection into diffusion models. In particular, when zooming in on the timestep embedding space, we find that different timesteps exhibit distinct representational capabilities that can encode side-channel information. Moreover, such encoded information can be utilized for attack and defense purposes through the scheduler interface. We present a theoretical analysis of timestep embeddings as position-encoding mappings and derive a mutual coherence evaluation that explains the separability of disjoint timestep intervals. Our findings reveal the diffusion model's timestep as a powerful side channel for carrying dedicated information, motivating new directions for adversarial generative modeling. The code is available at: https://github.com/OldDreamInWind/STE-Diffusion

[1]Department of Computer Science, University of Nevada Las Vegas, Las Vegas, USA. Correspondence to: Zuobin Xiong <zuobin.xiong@unlv.edu>.

*Proceedings of the 43rd International Conference on Machine Learning*, Seoul, South Korea. PMLR 306, 2026. Copyright 2026 by the author(s).

## 1. Introduction

Trained and fine-tuned on large-scale datasets (Schuhmann et al., 2022; 2021), diffusion models (DMs) (Ho et al., 2020; Song et al., 2021; Dhariwal & Nichol, 2021; Ho et al., 2022; Ho & Salimans, 2021; Rombach et al., 2022; Zhang & Chen, 2023) have been utilized as the standard backbone for high-fidelity content generation across images, video, audio, and natural language, achieving state-of-the-art generation quality through iterative denoising. Although early diffusion models, e.g., DDPM, require thousands of denoising steps, recent advances in sampling efficiency have reduced the number of sampling steps by one to two orders of magnitude (Lu et al., 2022; Zhao et al., 2023), accelerating real-world deployment across consumer and enterprise pipelines. However, this growing ubiquity of diffusion models raises urgent questions about safety, accountability, and provenance (Guo et al., 2025; Carlini et al., 2023; Duan et al., 2023; Truong et al., 2025), as the powerful AI tool enables precise generative control, which can be misused to produce harmful content (Zhang et al., 2024). Therefore, understanding the comprehensive threat surfaces and security implications in diffusion pipelines and how they can be subverted has become an important research topic.

Early works (Yu et al., 2023; Chen et al., 2025a) show that diffusion models can be used as particularly powerful steganographic carriers, in which auxiliary information is hidden within model parameters or latent representations via steganography (Younis et al., 2025; Sanjalawe et al., 2025) while preserving the perceptual fidelity of generated data. These findings suggest that diffusion models support both output-level and model-level secret channels to `embed information`. On the other hand, recent security studies demonstrate that diffusion models are vulnerable to a set of attacks (Chen et al., 2023; Shan et al., 2024; Chou et al., 2023a), especially backdoor attacks (Lin et al., 2026). In such a scenario, the attacker can implant backdoor malicious triggers into diffusion models based on steganography (i.e., `embedding malicious information`), so that crafted prompts can produce undesired outputs at inference time (Chou et al., 2023b; Zhai et al., 2023; Chen et al., 2025d;b). On the contrary, steganography can also serve as a watermark mechanism for model attribution and

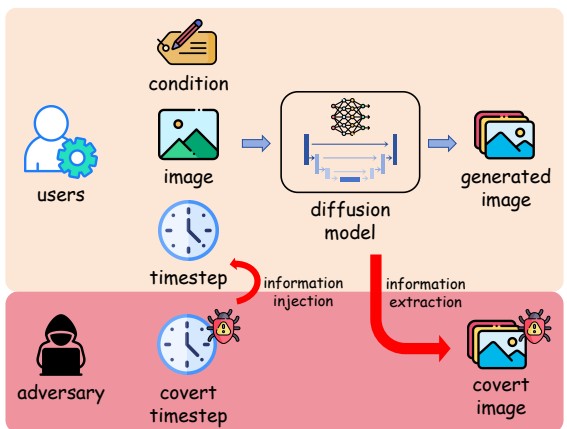

*Figure 1.* Adversaries can use the timestep as a secret channel, which can inject extra information.

content traceability by injecting signed embeddings and verifying their presence in the model output (Li et al., 2025; Wang et al., 2025a).

So far, the mainstream methods in information injection in diffusion models focused on conditions and output (e.g., generated images) aspects, while the intrinsic control over the timestep embedding of the model is overlooked, which could exhibit significant potential in different applications. This very research gap motivates us to study whether the temporal embeddings in diffusion models can function as a covert information channel, enabling isolated generative behaviors without modifying the observable sampling procedure.

Moreover, through literature review, we found that timestep security is a pressing and practical issue in diffusion models because they are inconspicuous yet essential. For instance, as shown in Fig. 1, an adversary can exploit the temporal embedding space by subtly encoding information in the timestep embeddings. Such malicious manipulation can be achieved through existing code poisoning attacks (Gokkaya et al., 2026; Wan et al., 2024), where the users mistakenly import disguised pipelines that are published by attackers. Therefore, these methods can become stealthier, leave fewer fingerprints, and easily evade defenses that monitor input/output spaces in traditional settings. Based on the application, this work introduces a novel information injection method, Shadow Timestep Embedding (**STE**), which operates on the temporal side-channel, introducing an invisible yet controllable timestep embedding interface.

Specifically, we find that modifying the timestep range in diffusion models can extend the embedding resolution, providing an unoccupied subspace for more information injection. Our analysis illustrates that the injected information can be malicious or legitimate, and this insight extends to broader security scenarios beyond steganography.

The contributions of this work are as follows.

- We propose Shadow Timestep Embedding (STE), a timestep-based information injection method for diffusion models, which uncovers an underutilized temporal channel to encode additional, controllable information.

- We demonstrate that the extended embedding space can introduce a dual-use security surface, e.g., STE can serve as both a covert attack injection method and a watermark verification tool.

- Through theoretical analysis, we prove the mutual coherence between different timesteps, revealing the fundamental reason that the embedding space can hold representation.

- Experiment results highlight that STE can inject auxiliary data distributions reliably while maintaining independence between the explicit and shadow manifolds.

## 2. Related works

**Steganography in Diffusion Models.** Recent work has explored diffusion models as powerful carriers for steganography, leveraging the denoising process to embed and recover hidden information. StegaDDPM (Peng et al., 2023) embed secret messages into the denoising trajectory or noise space, achieving high-capacity and visually imperceptible steganography. Training-free approaches such as CRoSS (Yu et al., 2023) introduce controllable and secure steganographic mechanisms by explicitly conditioning the diffusion process. Complementary to output-level hiding, DMIH (Chen et al., 2025a) demonstrates that diffusion models themselves can act as steganographic containers, embedding hidden image mappings directly into the learned score function.

However, prior approaches do not fully exploit the representational capacity of the temporal timestep embedding, which offers a covert and practical mechanism for information injection and extraction.

**Security of Diffusion Models.** The rapid deployment of diffusion models has raised concerns about their robustness and provenance. Backdoor attacks demonstrate that diffusion models can be maliciously fine-tuned to produce attacker-chosen outputs under specific triggers while behaving normally otherwise. VillanDiffusion (Chou et al., 2023b) provides a unified framework for both unconditional and conditional backdoors, revealing their persistence across different schedulers. Similarly, BadT2I (Zhai et al., 2023) embeds multi-modal triggers into text-to-image systems with minimal data poisoning. On the defensive side, concept erasure methods (Gandikota et al., 2023; Gong

et al., 2024; Chen et al., 2025c) attempt to remove harmful or copyrighted content via targeted parameter updates, whereas watermarking aims to establish content provenance. Specifically, Tree-Ring (Wen et al., 2023) encodes reversible frequency-domain signatures along the full sampling trajectory, and ROBIN (Huang et al., 2024) leverages adversarial optimization to embed robust, invisible watermarks aligned with diffusion dynamics.

Despite these advances, most security research focuses on data, prompts, or global model parameters, leaving the timestep embedding itself an under-examined dimension.

## 3. Method

### 3.1. Preliminaries

**Diffusion Models.** Diffusion models are a class of generative models that learn to synthesize data by inverting a gradual noising process. Given a clean sample $\mathbf{x}_0 \sim q(\mathbf{x}_0)$, the forward process progressively perturbs $\mathbf{x}_0$ through a sequence of Gaussian transitions:

$$q(\mathbf{x}_t \mid \mathbf{x}_{t-1}) = \mathcal{N}\left(\sqrt{1-\beta_t}\,\mathbf{x}_{t-1},\, \beta_t \mathbf{I}\right), \quad t = \{1, \ldots, T\}, \tag{1}$$

where $\{\beta_t\}_{t=1}^T$ is a pre-defined variance schedule controlling the noise magnitude at each timestep.

The generative process learns to invert this corruption by predicting the added noise $\boldsymbol{\epsilon}$ at each timestep. A neural network $\boldsymbol{\epsilon}_\theta(\mathbf{x}_t, t, \mathbf{c})$ parameterized by $\theta$ is trained to approximate the conditional mean of the reverse transition:

$$p_\theta(\mathbf{x}_{t-1} \mid \mathbf{x}_t) = \mathcal{N}\left(\boldsymbol{\mu}_\theta(\mathbf{x}_t, t, \mathbf{c}),\, \sigma_t^2 \mathbf{I}\right), \tag{2}$$

where $\mathbf{c}$ denotes optional conditioning (e.g., text or class label) and $t$ is represented via a *timestep embedding*. The standard training objective minimizes the denoising error between predicted and true noise:

$$\mathcal{L}_{\text{DM}}(\theta) = \mathbb{E}_{t, \mathbf{x}_t, \boldsymbol{\epsilon}}\left[\|\boldsymbol{\epsilon} - \boldsymbol{\epsilon}_\theta(\mathbf{x}_t, t, \mathbf{c})\|_2^2\right]. \tag{3}$$

Each discrete timestep $t$ (typically $t \in [0, 1000]$) is mapped to a continuous vector using sinusoidal or learned positional encoding, denoted $\mathbf{emb}_t = \Phi(t)$. This embedding acts as a global conditioning signal broadcast to every block of the denoising network, effectively coupling the scheduler's time dynamics with the model's internal representation. Our work builds on this observation and extends the embedding range beyond the conventional setup to explore the information capacity of the temporal subspace.

### 3.2. Shadow Timestep Embedding

Standard diffusion models operate over a fixed and compact timestep range $t \in [0, T_0]$, where each $t$ is mapped to a continuous feature vector through a learned or sinusoidal positional encoding. This design implicitly assumes that the entire temporal pathway is fully utilized during training.

However, for the positional encoding of timestep embeddings, the maximum period is typically set to 10,000, which means that a large portion of the timesteps (from 1,000 to 10,000) remain unused and only a limited fraction of the embedding spectrum is activated. This observation opens the possibility of constructing additional and functional independent temporal subspaces.

**Shadow Offsets.** Shadow Timestep Embedding (STE) extends the original timestep domain by introducing a temporally shifted set of indices via

$$T_n = T_0 + f_n, \quad f_n \geq 0, \tag{4}$$

where $f_n$ is the $n$-th offset of shadow timestep that maps the shadow interval $[0, T_0] \mapsto [f_n, T_n]$. While the scheduler continues to operate strictly over the standard timestep trajectory, the model receives the shifted index $t_{sn}$ instead of $t$, thereby projecting the computation into a new and well-separated embedding space $e_{t_{sn}}$ other than $e_t$:

$$e_t = \Phi(t), \quad e_{t_{sn}} = \Phi(t_{sn}), \tag{5}$$

where $\Phi$ is the embedding function and $t_{sn} \in [f_n, T_n]$ is the shadow timesteps. If $\Phi(t)$ is a smooth but nonlinear mapping, offsetting $t$ induces embedding vectors that are almost orthogonal to the standard range (see proof in Appendix). This separability enables the formation of a parallel shadow temporal manifold, which can encode data distributions that are not presented during explicit timesteps.

Importantly, STE is not tied to a specific discretization of the diffusion trajectory; instead, it operates on the timestep embedding function $\Phi(t)$ itself. In discrete-time diffusion models, the model receives timestep indices $t \in \{1, \ldots, T_0\}$, whereas in continuous-time generative models, such as flow matching, the time variable can be sampled from a continuous interval, e.g., $t \sim \mathcal{U}(0, 1)$. STE can therefore be generalized by applying a continuous offset transformation, such as $t_s = t + \delta$, and feeding the shifted embedding $\Phi(t_s)$ to the denoising or vector-field network. As long as $\Phi(\cdot)$ is smooth, as in sinusoidal or learned positional encodings, sufficiently separated temporal intervals retain low mutual coherence, thereby preserving the separability between explicit and shadow distributions. This indicates that STE provides an extensible temporal channel for both discrete- and continuous-time generative frameworks.

**Learning Independent Data Distributions.** A key property of STE is that each offset $f_n$ can be associated to a distinct dataset. Samples associated with shadow timesteps $t_{sn}$ push the model to learn a distribution $\mathcal{D}_{sn}$ that is independent of the standard data distribution $\mathcal{D}_0$ learned under $t \in [0, T_0]$. STE uses different offsets to construct disjoint temporal bands, allowing each band to support a separate generative behavior.:

$$t \in [0, T_0] \longrightarrow \mathcal{D}_0, \quad t_{sn} \in [f_n, T_n] \longrightarrow \mathcal{D}_{sn}. \tag{6}$$

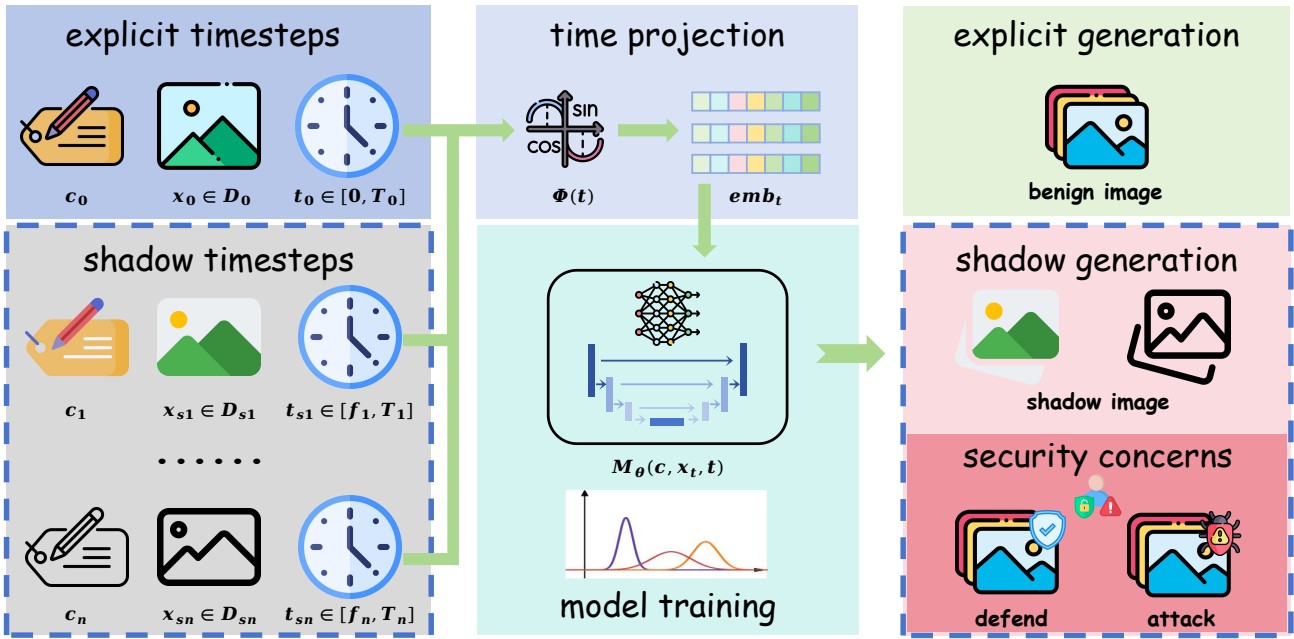

*Figure 2.* Architecture of Shadow Timestep Embedding (STE). The framework extends conventional diffusion training by introducing shadow timesteps, which parallel explicit timesteps but occupy disjoint temporal intervals. Each shadow timestep subset $t_{sn}$ is associated with different data distributions $D_{sn}$. During time projection, timesteps are encoded via sinusoidal embeddings and fed into the diffusion model for joint training. At inference, the model can generate either explicit or shadow images. Such multi-space learning enables new capabilities but also raises security concerns, as shadow subspaces can be exploited for covert defend or attack.

Thus, by substituting the shadow timestep $t_{sn}$ into the standard diffusion objective in Eq. 3, we obtain the STE-specific loss function:

$$\mathcal{L}_{\text{STE}}(\theta) = \mathbb{E}_{t,\mathbf{x}_t,\boldsymbol{\epsilon}}\big[\|\boldsymbol{\epsilon} - \boldsymbol{\epsilon}_\theta(\mathbf{x}_t,t,\mathbf{c})\|_2^2\big]$$
$$+ \mathbb{E}_{t_{sn},\mathbf{x}_{t_{sn}},\boldsymbol{\epsilon}}\big[\|\boldsymbol{\epsilon} - \boldsymbol{\epsilon}_\theta(\mathbf{x}_{t_{sn}},t_{sn},\mathbf{c}_{sn})\|_2^2\big], \quad (7)$$

where $\mathbf{c}_{sn}$ is the condition of the specific shadow dataset.

Significantly, during the shadow timestep shift process, the scheduler needs no modification. From the perspective of the user (i.e., the model owner or victim), generation proceeds with the usual sequence $t = [T_0, \ldots, 0]$. The model, however, interprets the timestep $t$ as $t_{sn} \in [T_n, \ldots, f_n]$, enabling shadow generation that follows the same numerical integration path as explicit generation but activates an entirely different learned distribution. As shown in Fig. 2, the offset serves as a "temporal key", a mechanism that switches the explicit generative behavior to shadow behavior without altering scheduler dynamics.

**Security Implications.** The ability to conceal a separate distribution within a shadow timestep interval has profound security implications, including for attack and defense.

Attack Perspective. Attackers can use STE for a stealthy information injection attack. When $\mathcal{D}_{sn}$ contains poisoned data, the offset $f_n$ itself becomes the trigger. The model behaves normally for $t \in [0, T_0]$ but produces hidden information when exposed to $t_{sn}$. This attack pathway is covert

because both the scheduler and the model interface pipeline remain unchanged.

Defense Perspective. Defender can use STE as an ownership verification mechanism. When $\mathcal{D}_{sn}$ consists of images containing structured watermark signals, STE becomes a high-fidelity, invisible watermarking scheme. The watermark is activated only by stepping into the shadow interval, making it robust against common post-processing.

In summary, STE introduces an extensible temporal subspace that simultaneously (i) preserves compatibility with existing schedulers, (ii) supports independent generative behavior, and (iii) opens a new and largely unexplored security surface, either as a vulnerability for adversaries or as a useful primitive for secure provenance.

### 3.3. Mutual Coherence Between Temporal Intervals

The preceding section introduces an analysis of the difference between standard and shadow timestep embeddings. If we try to inject additional information into this extended space, the natural question arises: *are these shadow embeddings separable enough to encode distinct information?*

To address this, we analyze the **mutual coherence** between embeddings from different temporal intervals and derive the following theorem.

**Theorem 3.1** (Mutual Coherence of STE)**.** *Let $\Phi(t) \in \mathbb{R}^d$*

denote the sinusoidal timestep embedding used in diffusion models:

$$\Phi(t) = \big[\sin(\omega_1 t), \cos(\omega_1 t), \ldots, \sin(\omega_m t), \cos(\omega_m t)\big], \quad (8)$$

where $d = 2m$, and the frequencies follow a geometric progression $\omega_i = \exp\left(-\frac{\log(t_p)}{m - t_d}(i - 1)\right)$, where $t_p$ is the maximum time period and $t_d$ is the downscale frequency shift time.

Then, (1) for two timesteps $t, s \in \mathbb{R}$, the cosine similarity between embeddings is

$$k(t, s) = \frac{1}{m} \sum_{i=1}^{m} \cos\big(\omega_i(t - s)\big), \quad (9)$$

and (2) the mutual coherence between two disjoint temporal intervals $I_0 = [0, t_0)$ and $I_1 = [t_0, t_1)$ is upper bounded by

$$\mu = \sup_{\Delta \in [t_0, t_1]} \left| \frac{1}{m} \sum_{i=1}^{m} \cos(\omega_i(t - s)) \right|. \quad (10)$$

*Proof.* We refer the readers to the Appendix for the complete proof. □

In Eq. (10), $\mu$ indicates that embeddings from $I_0$ and $I_1$ are nearly orthogonal, hence linearly separable in the embedding space. This separability is crucial for STE because it ensures that extending timesteps into a new interval does not collapse into the same feature manifold, but instead provides an independent channel for encoding auxiliary distributions.

**Empirical Observations.** To demonstrate this property, we visualize empirical statistics of timestep embeddings, as shown in Figure 3. We measure $\mu$ in Fig. 3a across the full timestep range and observe that when $t - s \geq 1000$, the coherence remains very small and swings around 0.1, indicating that embeddings between the normal and shadow intervals are nearly orthogonal. This validates that extended timesteps form a distinguishable subspace suitable for additional information channels. For $t - s < 1000$, the performance of the model will degrade, which is caused by the region overlap of timestep embedding and the difference in noise level of timestep $t$. Possible side effects of this setting are discussed in Section 4.6.

A heatmap of 128-dimensional embeddings in Fig. 3b illustrates distinct value patterns across timesteps, especially beyond the original training range of 1000 steps. Such spatially varying intensity patterns verifies that extended timesteps yield significantly different embedding trajectories from their normal counterparts.

Together, these results confirm that the timestep embedding space is rich enough to support multiple, partially independent subspaces. By extending the temporal domain and leveraging its low mutual coherence, we can strategically

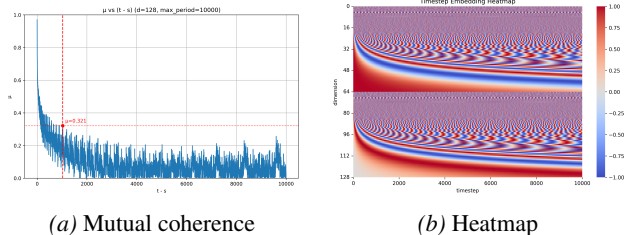

*(a)* Mutual coherence   *(b)* Heatmap

*Figure 3.* Experiments on separability of time embeddings

introduce *shadow timesteps* as an auxiliary encoding mechanism without interfering with the model's original dynamics. This property underpins the feasibility of injecting new data distributions or watermark signals purely through the temporal conditioning pathway.

## 4. Experiments

### 4.1. Experiment Setup

We evaluate Shadow Timestep Embedding (STE) across three dimensions: (i) **general performance**, measuring whether extending the temporal embedding space affects the model's core generative quality and the extraction accuracy for the hidden distribution, (ii) **attack performance**, examining STE as a covert information injection attack mechanism, and (iii) **defend performance**, assessing the performance of STE as a watermark generator. Our experiments are designed to fully characterize both the utility and representational capability of the timestep domain.

**Baselines.** We benchmark STE against three categories of baselines: (1) **Diffusion scheduler baselines:** DDPM, DDIM, and DPM-Solver (Ho et al., 2020; Song et al., 2021; Lu et al., 2022), representing standard probabilistic, deterministic, and ODE-based sampling behaviors. (2) **Backdoor attack baselines:** VillanDiffusion and BadDiffusion (Chou et al., 2023b;a), two state-of-the-art methods showing that diffusion models can be backdoored via multimodal or image–condition triggers. These baselines establish the achievable attack power when the adversary operates in pixel space or conditioning space. Our STE extends this comparison to the timestep domain. (3) **Watermark baselines:** Tree-Ring, ROBIN, and SleeperMark (Wen et al., 2023; Huang et al., 2024; Wang et al., 2025b), representing reversible trajectory-based watermarks and adversarially optimized robust watermarks, respectively. By comparing STE-based watermark encoding to existing pixel- and frequency-space watermarking, we quantify whether the temporal channel can afford additional robustness or stealth.

**Metrics.** To comprehensively assess STE, we adopt the following evaluation metrics: (1) **FID**, standard generative quality metric evaluating the distributional distance between generated and real images. (2) **Accuracy (ACC)**, we

*Table 1.* General performance of STE under different settings. The timestep type is ordered as [CIFAR-10,MNIST,Fashion-MNIST]. Here, $E$ denotes the explicit timestep interval, $S$ denotes a shadow timestep interval, and $-$ denotes an unassigned interval. The baseline row summarizes three separately trained single-dataset DDPMs. In STE settings, CIFAR-10 is assigned to the explicit timestep interval, while MNIST and Fashion-MNIST are assigned to shadow intervals with offsets 1000 and 2000, respectively. Best values are bolded only among assigned configurations.

| Setting | Timestep Type | CIFAR-10 | | | MNIST | | | Fashion-MNIST | | |
|---|---|---|---|---|---|---|---|---|---|---|
| | | FID↓ | ACC↑ | ER↓ | FID↓ | ACC↑ | ER↓ | FID↓ | ACC↑ | ER↓ |
| Single-dataset | Baseline | 24.38 | 73.41% | **8.02%** | 1.59 | **98.33%** | **10.58%** | 12.16 | 88.12% | 8.63% |
| STE | $[E, S, -]$ | 22.20 | 73.35% | 8.69% | **1.18** | 97.55% | 10.75% | 392.79[†] | 10.41%[†] | 10.13%[†] |
| STE | $[E, -, S]$ | 23.32 | 75.03% | 8.85% | 351.23[†] | 9.63%[†] | 10.06%[†] | **3.31** | 87.96% | 8.99% |
| STE | $[E, S, S]$ | **21.82** | **75.65%** | 9.17% | 1.85 | 97.77% | 10.92% | 6.56 | **88.76%** | **8.14%** |

[†] denotes an inactive interval with no assigned dataset; these entries are reported to test distribution isolation.

measure whether generated samples preserve correct class semantics. This also enables quantifying whether shadow-timestep generation unintentionally leaks into unintended distributions. (3) **Exposure Rate (ER)**, a metric introduced to quantify unintended dataset leakage. ER is defined as the mean classification accuracy of a generated set when evaluated by classifiers trained on other datasets. Higher ER indicates that the generated samples are unintentionally exposed to an undesired timestep range. (4) **Attack Success Rate (ASR)**, it measures the fraction of generations that successfully extract the attacker-chosen target distribution when triggered by a shadow timestep offset. (5) **Watermark Detection Accuracy**, for watermark experiments, we measure the proportion of detection accuracy by a watermark detector.

**Base Model and Training Configuration.** Unless otherwise specified, all experiments use the DDPM architecture and training pipeline as described in the original paper. STE is implemented by augmenting the timestep range beyond the standard $T_0 = 1000$ interval to create shadow intervals $t_{sn} \in [f_n, f_n + T_0]$, where each offset $f_n$ is set as an integer multiple of 1000. Unless explicitly stated, each shadow interval corresponds to one additional dataset or security-related data distribution.

All models are trained for 100 epochs with a learning rate of $2 \times 10^{-4}$. Training is performed on NVIDIA L40S GPUs. This configuration is used consistently across general evaluation, backdoor experiments, and watermark experiments to enable fair comparison across settings.

### 4.2. General STE Performance

We first evaluate whether STE can incorporate multiple datasets with different timestep intervals while preserving generation quality and distribution separability, as shown in Table 1. Table 1 reports the performance on CIFAR-10, MNIST, and Fashion-MNIST under several timestep-assignment settings. The first row corresponds to single-dataset baselines, where three DDPMs are trained separately

on CIFAR-10, MNIST, and Fashion-MNIST. The remaining rows correspond to STE models, where CIFAR-10 is assigned to the explicit timestep interval, and MNIST or Fashion-MNIST can be assigned to shadow timestep intervals.

**Generation Quality under Different Settings.** The results show that adding shadow timestep intervals does not lead to consistent degradation in generation quality. Instead, the behavior is non-monotonic across datasets and metrics. For CIFAR-10, all STE settings improve FID relative to the single-dataset baseline, reducing it from 24.38 to 22.20, 23.32, and 21.82. The full STE setting $[E, S, S]$ achieves the best CIFAR-10 FID and ACC, with FID $= 21.82$ and ACC $= 75.65\%$. For MNIST, the $[E, S, -]$ setting achieves the best FID of 1.18, while the single-dataset baseline retains the highest ACC. For Fashion-MNIST, the $[E, -, S]$ setting achieves the best FID, whereas the full $[E, S, S]$ setting achieves the highest ACC and the lowest ER. These results indicate that assigning additional datasets to shadow timestep intervals does not universally harm sample quality and can even improve certain metrics.

**Distribution Isolation.** The entries marked with † correspond to unassigned intervals. For example, Fashion-MNIST is unassigned in the $[E, S, -]$ setting, and MNIST is unassigned in the $[E, -, S]$ setting. These entries are not used to compare generation quality; instead, they serve as negative controls for distribution isolation. The very high FID values and near-chance ACC values in these unassigned intervals indicate that the model does not accidentally generate samples from a dataset that has not been assigned to that timestep interval. This supports the intended isolation property of STE: assigned explicit or shadow intervals learn their corresponding distributions, while unassigned intervals do not collapse into those of other datasets.

**Trade-off in Multi-distribution Learning.** Although STE preserves overall generation quality, some dataset-specific metrics still vary across settings. For instance, CIFAR-10 obtains its best FID and ACC under $[E, S, S]$, but its ER

*Table 2.* Comparison of Different Attack Methods on Diffusion Models. The STE-explicit uses a clean dataset. The STE-shadow binds the covert dataset.

| Method | CIFAR-10 | | Celeba-HQ | |
|---|---|---|---|---|
| | FID↓ | ASR↑ | FID↓ | ASR↑ |
| VillanDiffusion | 25.66 | 96.2% | 6.53 | 97.7% |
| BadDiffusion | 22.53 | **99.5%** | 7.65 | 98.7% |
| STE-explicit | **21.82** | 0.2% | **6.23** | 0.4% |
| STE-shadow | 22.07 | 99.2% | 6.78 | **98.8%** |

*Table 3.* Robustness comparison of different watermarking methods on diffusion models under common image distortions.

| Method | Blur | Noise | JPEG | Bright | Crop | Avg |
|---|---|---|---|---|---|---|
| Tree-Ring | 0.98 | 0.98 | 0.94 | 0.86 | 0.99 | 0.95 |
| ROBIN | **0.99** | **0.99** | **0.97** | 0.95 | **1.00** | **0.98** |
| SleeperMark | 0.97 | 0.85 | 0.96 | **0.96** | - | 0.94 |
| STE | 0.96 | 0.98 | 0.96 | 0.85 | 0.71 | 0.89 |

is slightly higher than that of the single-dataset baseline. Similarly, MNIST obtains the best FID under $[E, S, -]$, but not under the full $[E, S, S]$ setting. This suggests that incorporating multiple heterogeneous datasets introduces a trade-off between modeling several timestep-separated distributions and optimizing dataset-specific fidelity.

Overall, Table 1 shows that STE does not simply degrade as more datasets are assigned to shadow intervals. Its performance reflects a balance between multi-distribution generalization, sample quality, and distribution isolation.

### 4.3. STE as Security Channel

Beyond improving temporal representation, STE exposes a dual-use temporal pathway that can function as an independent security channel. Because explicit and shadow timesteps form nearly orthogonal temporal manifolds, the model can simultaneously support a clean explicit distribution and a security-relevant shadow distribution. We explore two applications of this property: (i) STE as an information injection attacker and (ii) STE as a watermark verification.

**Use STE as an Information Injection Attacker.** To evaluate whether STE can be exploited as a covert backdoor attack mechanism, we bind the explicit timestep interval to a clean dataset and the shadow timestep interval to a poisoned dataset with a 5% poisoning rate. The model therefore learns two isolated behaviors: benign generation under normal timesteps, and adversarial behavior when the shifted shadow timesteps are activated. The explicit branch remains visually faithful to the clean CelebA-HQ distribution, while the shadow branch (STE attack) reliably generates the attacker's target pattern, a "bug" backdoor sample. Table 2 shows that STE-shadow achieves an Attack Success Rate of 99.2% on CIFAR-10 and 98.8% on CelebA-HQ. In contrast, STE-explicit maintains extremely low ASR (0.2% and

0.4%) while achieving the best FID among all methods. The explicit timesteps pathway preserves the best utility and quality for non-poisoned data generation, while the attack is fully contained within the shadow interval. Such separation makes the attack more stealthy because neither pixel statistics nor model weights exhibit abnormalities.

**Use STE as Watermark Verification.** We then treat the STE shadow timesteps as an in-process watermark generator that binds them to a dataset augmented with a visible shield watermark. The model thus acquires the ability to generate either clean images (via explicit timesteps) or watermarked images (via shadow timesteps). Table 3 summarizes robustness under common distortions for different watermark attacks. STE maintains high watermark detectability under blur, noise, and JPEG compression, competitive with baselines. However, because the watermark in our study is explicit and pixel-visible, STE is vulnerable to brightening and cropping attacks, dropping to 0.85 and 0.71, respectively. Although the current pixel-level watermark vulnerability stems from the explicit nature of our chosen watermark, STE provides a dual-mode generation mechanism that allows the user to freely toggle between clean and watermarked outputs by selecting normal and shadow timesteps. STE uniquely enables this temporal disentanglement and is not offered by existing pixel-space watermarking methods. In our future work, we will study more complex watermark patterns, such as frequency- or latent-level patterns, to improve the watermarking mechanism.

Across both applications, STE demonstrates that timestep embedding is a powerful and flexible security channel. *Shadow timesteps act as temporal keys that selectively inject information while preserving standard model performance.*

### 4.4. Extensibility to Flow Matching Models

We further evaluate whether STE can be extended beyond DDPM by applying it to flow matching models, which provide a continuous-time generative formulation. As shown in Table 4, STE exhibits comparable behavior under DDPM and flow matching. In the STE setting, flow matching improves the CIFAR-10 FID from 22.20 to 19.44 while maintaining similar ACC and ER, and it achieves comparable ACC and ER on MNIST despite a higher FID. This suggests that the observed FID differences mainly reflect the modeling characteristics of the underlying generative framework rather than a failure of STE. More importantly, the consistency of ACC and ER indicates that the explicit and shadow timestep intervals remain well separated in both DDPM and flow matching.

The reason is that STE operates on the timestep embedding space rather than on a specific DDPM sampling trajectory. Its effectiveness relies on assigning different distributions to separated regions of the timestep embedding function. The

*Table 4.* The performance of STE on DDPM and Flow Matching.

| Setting | Dataset | FID (DDPM)↓ | FID (Flow)↓ | ACC (DDPM)↑ | ACC (Flow)↑ | ER (DDPM)↓ | ER (Flow)↓ |
|---------|---------|-------------|-------------|-------------|-------------|------------|------------|
| Clean | CIFAR-10 (single) | 24.38 | 20.45 (−3.93) | 73.41% | 73.56% (+0.15) | 5.39% | 6.43% (+1.04) |
| Clean | MNIST (single) | 1.59 | 3.22 (+1.63) | 98.33% | 98.80% (+0.47) | 9.71% | 8.73% (−0.98) |
| STE | CIFAR-10 (Explicit) | 22.20 | 19.44 (−2.76) | 73.35% | 73.22% (−0.13) | 6.48% | 6.05% (−0.43) |
| STE | MNIST (shadow) | 1.18 | 3.25 (+2.07) | 97.55% | 97.24% (−0.31) | 9.86% | 9.46% (−0.40) |

*Table 5.* FID performance of STE with different schedulers and sampling steps.

| Scheduler | Steps | CIFAR-10 | MNIST | Fashion-MNIST |
|-----------|-------|----------|-------|---------------|
| DDPM | 1000 | **21.82** | **1.85** | **6.56** |
| DDIM | 50 | 22.27 | 3.40 | 7.28 |
| DPM-Solver | 20 | 37.88 | 16.14 | 9.97 |

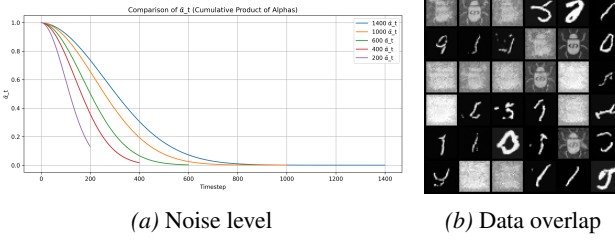

*(a)* Noise level        *(b)* Data overlap

*Figure 4.* Analysis of scheduler with different training timesteps.

flow matching results in Table 4 support this extensibility and suggest that STE is not limited to discrete-time DDPMs.

### 4.5. Impact of Schedulers

Table 5 evaluates STE under different schedulers using the $[E, S, S]$ configuration from Table 1 as the baseline. Different schedulers use various sampling timesteps. DDPM achieves the most stable performance across CIFAR-10, MNIST, and Fashion-MNIST, confirming its compatibility with STE's extended temporal domain. DDIM shows slightly higher FID but remains competitive while offering significantly faster sampling, representing a good balance between efficiency and quality. DPM-Solver exhibits substantially degraded performance across all datasets. Overall, the results indicate that STE is most robust when paired with schedulers that preserve DDPM-like schedulers.

### 4.6. Impact of Training Timesteps

To investigate whether the alternative timestep-embedding design could work, we train models by modifying the scheduler's timestep range while keeping the model's embedding unchanged. Figure 4a illustrates how different schedulers inject noise at specific intensities when operating on different training timestep ranges. Although these schedulers follow the same nominal trajectory, the actual noise levels vary considerably, forcing the model to learn multiple noise-to-data mappings for the same timestep index. This leads

to the failure mode visualized in Fig. 4b. Because multiple training datasets are associated with the same timestep, each is mapped to a different noise magnitude determined by the scheduler, leading to strong gradient conflict. Consequently, the learned distributions across different datasets begin to overlap, preventing the model from establishing clear temporal separation.

### 4.7. Potential Defense Mechanisms

STE introduces a previously overlooked security surface: the temporal pathway in diffusion models. To address this concern, we outline potential defense directions motivated by our analysis. STE exploits a mismatch between the scheduler trajectory and the timestep values actually fed to the model. A defense strategy can verify the consistency between the scheduler-issued timestep $t$ and the value the denoiser receives. A systematic offset indicates a potential temporal manipulation and can be flagged if detected. Also, the embedding vectors for shadow timesteps lie in regions with low mutual coherence relative to the standard timestep interval. A defense module can exploit this by monitoring the distribution of timestep embeddings. A lightweight anomaly-detection mechanism can detect deviations from the manifold it provides. However, a comprehensive defense against STE-based attacks remains an open challenge, requiring further work on secure scheduler design and temporal provenance.

## 5. Conclusion

In this work, we introduced STE, a mechanism that explores the temporal dimension of diffusion models and reveals its untapped representational capacity. By extending the timestep domain beyond the standard training range, STE constructs parallel temporal manifolds that can encode information or independent data distributions. Our analysis shows that these shadow timesteps form nearly orthogonal embedding regions, thereby enabling STE to serve as a powerful channel for information injection. Extensive experiments confirm that STE preserves generation fidelity, supports learning from isolated distributions, and enables security applications with high success rates. These findings reveal the timestep embedding pathway as a critical yet previously overlooked security surface in diffusion models, motivating new directions for secure generative modeling.

## Acknowledge

This work was supported by the National Science Foundation under grants No. 2429960, 2434899, and 2548041, and the Institute of Information & communications Technology Planning & Evaluation (IITP) grant funded by the Korea government (MSIT) (No. RS-2024-00431388, the Global Research Support Program in the Digital Field program).

## Impact Statement

This paper presents work aimed at advancing the field of Machine Learning. There are many potential societal consequences of our work, none of which we feel must be specifically highlighted here.

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

## A. Theory Analysis

**Theorem A.1** (Mutual Coherence of STE). *Let $\Phi(t) \in \mathbb{R}^d$ denote the sinusoidal timestep embedding used in diffusion models:*

$$\Phi(t) = \big[\sin(\omega_1 t),\, \cos(\omega_1 t),\, \ldots,\, \sin(\omega_m t),\, \cos(\omega_m t)\big], \tag{11}$$

*where $d = 2m$, and the frequencies follow a geometric progression*

$$\omega_i = \exp\left(-\frac{\log(t_p)}{m - t_d}(i - 1)\right), \tag{12}$$

*where $t_p$ is the maximum time period and $t_d$ is the downscale frequency shift time.*

*For two timesteps $t, s \in \mathbb{R}$, the similarity between embeddings is*

$$k(t, s) = \frac{\langle \Phi(t), \Phi(s) \rangle}{\|\Phi(t)\|\|\Phi(s)\|} = \frac{1}{m}\sum_{i=1}^{m}\cos\big(\omega_i(t - s)\big). \tag{13}$$

*Proof.* For two timesteps $t, s \in \mathbb{R}$, the inner production between embeddings is

$$K(t, s) = \langle \Phi(t), \Phi(s) \rangle. \tag{14}$$

Applying the trigonometric identity $\sin A \sin B + \cos A \cos B = \cos(A - B)$, we obtain

$$\begin{aligned}
K(t, s) &= \sum_{i=1}^{m}\big[\sin(\omega_i t)\sin(\omega_i s) + \cos(\omega_i t)\cos(\omega_i s)\big] \\
&= \sum_{i=1}^{m}\cos\big(\omega_i(t - s)\big) \\
&= K(|t - s|).
\end{aligned} \tag{15}$$

Thus, the embedding implicitly defines a translation-invariant kernel over the time axis, whose similarity depends solely on the temporal difference $\Delta = t - s$.

Because each embedding component satisfies $\sin^2(\omega_i t) + \cos^2(\omega_i t)$, the squared norm of $\Phi(t)$ is

$$\|\Phi(t)\|^2 = m, \quad \text{independent of } t. \tag{16}$$

The normalized kernel (cosine similarity) is therefore

$$k(t, s) = \frac{K(t, s)}{\|\Phi(t)\|\|\Phi(s)\|} = \frac{1}{m}\sum_{i=1}^{m}\cos\big(\omega_i(t - s)\big). \tag{17}$$

We refer to $k(t, s)$ as the *normalized timestep kernel*. $\qquad\square$

**Definition A.2** (Mutual Coherence Between Temporal Intervals). Consider two disjoint temporal intervals $I_0 = [0, t_0)$ and $I_1 = [t_0, t_1)$. The **mutual coherence** between their embeddings is defined as

$$\begin{aligned}
\mu &= \sup_{t \in I_0,\, s \in I_1} |k(t, s)| \\
&= \sup_{\Delta \in [t_0, t_1]} \left|\frac{1}{m}\sum_{i=1}^{m}\cos(\omega_i(t - s))\right|.
\end{aligned} \tag{18}$$

A small $\mu$ indicates that embeddings from $I_0$ and $I_1$ are nearly orthogonal, hence linearly separable in the embedding space.

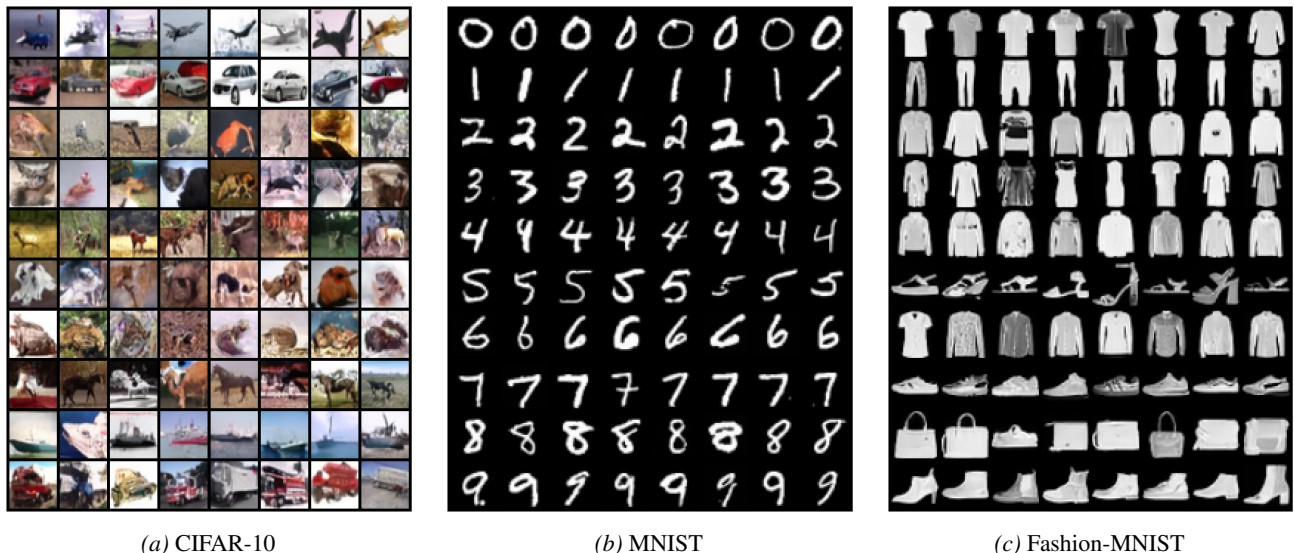

*(a)* CIFAR-10        *(b)* MNIST        *(c)* Fashion-MNIST

*Figure B1.* Visualization result of STE on DDPM model with CIFAR-10, MNIST, and Fashion-MNIST dataset. CIFAR-10 is on the explicit timesteps. MNIST is on the shadow timesteps with a 1000 offset. Fashion-MNIST is on the shadow timesteps with a 2000 offset.

## B. Additional Experiments

In this section, we provide additional qualitative results that complement the quantitative evaluation in the main paper and further illustrate the flexibility of Shadow Timestep Embedding (STE).

First, we visualize STE on the DDPM backbone trained jointly on CIFAR-10, MNIST, and Fashion-MNIST. In Figure B1a, CIFAR-10 is bound to the explicit timestep interval and produces diverse natural images with no visible degradation in visual quality. Figures B1b and B1c then assign MNIST and Fashion-MNIST to shadow timestep intervals with offsets of 1000 and 2000, respectively. The generated digits and clothing items remain sharp and class-consistent, indicating that the model can successfully multiplex multiple datasets across disjoint timestep ranges while maintaining high-fidelity samples for each domain.

Next, we investigate STE on the CelebA dataset under both attack and watermarking configurations. In the attack setting (Figure B2), sampling from explicit timesteps yields benign face images, whereas sampling from the corresponding shadow timesteps produces a targeted bug pattern. This demonstrates that STE can realize a distribution-level backdoor that is only activated when the sampler follows the shadow schedule, while normal usage remains unaffected. In the watermark setting (Figure B3), explicit timesteps generate standard faces, while shadow timesteps reproduce a protected distribution of faces, showing that STE can also be used to embed ownership information into the model in a way that is decoupled from the normal sampling procedure.

Finally, we extend our study to a large-scale text-to-image model, Stable Diffusion v1.5, on the COCO dataset. Using the same prompt (*"A dinner plate has a lemon wedge garnishment. A bug image."*), explicit timesteps produce a faithful, realistic interpretation of the text, whereas shadow timesteps yield images that are dominated by the STE-injected target pattern (Figure B4). These results confirm that STE naturally scales to modern text-to-image pipelines and can inject or retrieve information in the generation process without modifying the visible architecture or training objective beyond the altered timestep allocation.

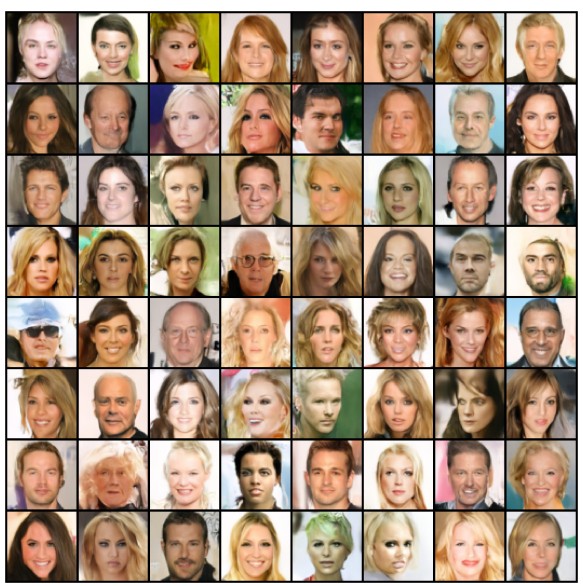 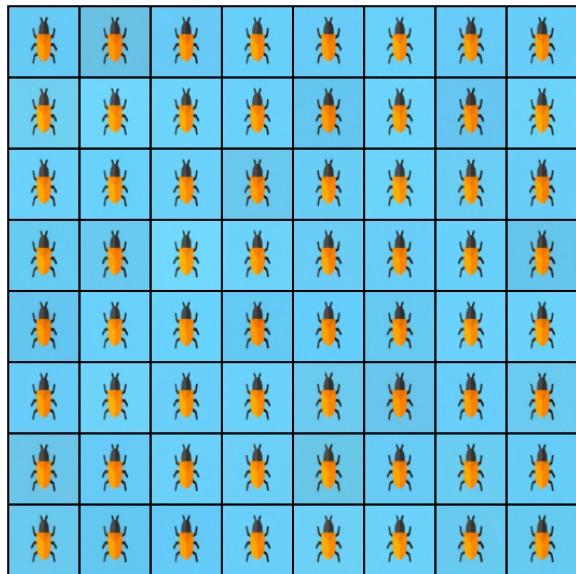

*(a)* Explicit Timesteps        *(b)* Shadow Timesteps

*Figure B2.* Visualization result of STE on DDPM model with Celeba dataset in attack setting. The left part is the normal generation results. The right part is the target generation results.

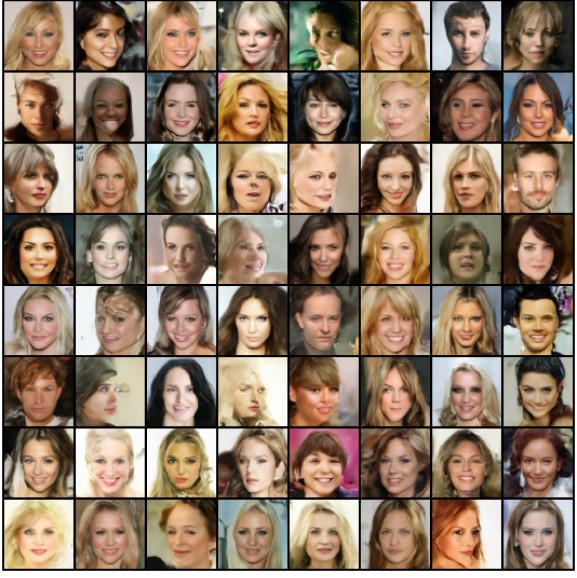

*(a)* Explicit Timesteps        *(b)* Shadow Timesteps

*Figure B3.* Visualization result of STE on DDPM model with Celeba dataset in watermark setting. The left part is the normal generation results. The right part is the protected generation results.

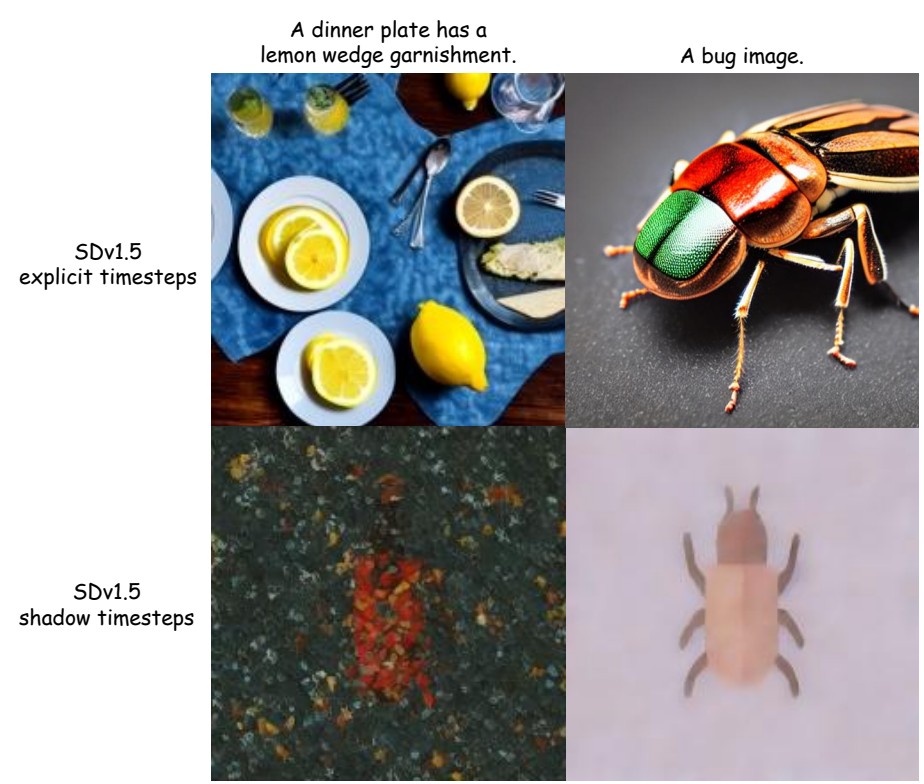

*Figure B4.* Visualization result of STE on Stable Diffusion Model and COCO dataset.

