# OpenReview forum: "Watch Your Step: Information Injection in Diffusion Models via Shadow Timestep Embedding"
_ICML.cc/2026/Conference — ICML 2026 regular_

### Official Review · Reviewer_tHbE · 2026-03-03

**Soundness:** 3
**Presentation:** 3
**Significance:** 3
**Originality:** 3
**Overall Recommendation:** 4
**Confidence:** 3

**Summary:**

This paper introduces Shadow Timestep Embedding (STE), a novel information injection mechanism for diffusion models that exploits underutilized temporal subspaces. The authors theoretically demonstrate that timestep embeddings from different temporal intervals exhibit extremely low mutual coherence, ensuring linear separability between data distributions. The effectiveness of this method is validated through extensive experiments in scenarios such as backdoor attacks and watermark protection

**Compliance With Llm Reviewing Policy:**

Affirmed.

**Final Justification:**

My final recommendation remains 4 (Weak Accept).

The paper is technically sound, reasonably novel, and clearly presented. My main concern was the lack of sufficiently convincing results in reduced-step or more continuous-time sampling settings. The rebuttal was helpful, but it did not fully address this point, since stronger evidence in these settings is still missing. Overall, my assessment is unchanged: this is a solid paper with clear merit, but with some limitations in evaluation scope, so I am maintaining my original score.

**Key Questions For Authors:**

- How does this method perform when the number of inference steps is small? Would a reduction in sampling steps lead to a failure in extracting the injected information, particularly for recent one-step or few-step generation methods?
- Based on the results in Table 1, the generation quality appears to degrade after incorporating three datasets. Could you provide a more detailed discussion on the relationship between model fidelity (FID/ACC) and the number or size of the injected datasets?

**Limitations:**

yes

**Strengths And Weaknesses:**

### Strengths
- The paper is clearly written and easy to understand.
- The paper is the first to propose using the timestep as a covert information channel for information injection. It provides a rigorous theoretical verification showing that distinct data distributions can be injected into different temporal intervals with minimal correlation.
### Weaknesses
- For information injection, the current approach requires re-training or fine-tuning the entire model whenever a new dataset is added. This process is significantly time-consuming.

---

> ### Author Rebuttal · Authors · 2026-03-31
>
> ## Response to Reviewer tHbE.
>
> We thank the reviewer for the positive feedback on the clarity and theoretical contributions of our work. We are encouraged that the reviewer recognizes the novelty of exploring timestep embeddings and the theoretical analysis in this domain.
>
> Below, we address the key questions.
>
> **Q1**: Performance under a small number of inference steps (few-step / one-step generation)
>
> We evaluate STE under different samplers and sampling steps in the main paper **Table 4**, with 3 different schedulers:
>
> - **DDPM (1000 steps)**
> - **DDIM (50 steps)**
> - **DPM-Solver (20 steps)**
>
> We observe that:
> - **DDIM (few-step)** achieves performance comparable to DDPM, indicating that STE remains effective under reduced sampling steps.
> - **DPM-Solver (very few steps)** shows noticeable degradation.
>
> We attribute this to a **mismatch between training and sampling paradigms**. Our base model is trained in a discrete DDPM framework, whereas DPM-Solver is designed for continuous-time ODE-based sampling, introducing an inconsistency between training and inference.
>
> Importantly, STE itself operates on the timestep embedding space, not on the sampling trajectory. Therefore, its effectiveness primarily depends on whether the sampling process can correctly traverse the intended timestep interval. In principle, if STE is applied to continuous-time trained models (e.g., SDE-based or flow-based models), we expect better compatibility with few-step or one-step samplers. This is partially supported by our experiments on flow-based models, as shown in the following table, and we will further investigate this direction in future work.
>
> Table R2 summarizes the comparison between DDPM and flow-based models:
>
> *Table R2: The performance of STE on DDPM and Flow Matching.*
>
> | Setting | Dataset | FID (DDPM) ↓| FID (Flow) ↓| ACC (DDPM)↑ | ACC (Flow) ↑ | ER (DDPM)↓ | ER (Flow) ↓|
> |------|--------|:----------:|:----------:|:----------:|:----------:|:---------:|:---------:|
> | Clean | cifar-10 (single) | 24.38 | 20.45 (-3.93) | 73.41% | 73.56% (+0.15) | 5.39% | 6.43% (+1.04) |
> | Clean | mnist (single) | 1.59 | 3.22 (+1.63) | 98.33% | 98.80% (+0.47) | 9.71% | 8.73% (-0.98) |
> | STE | cifar-10 (Explicit) | 22.20 | 19.44 (-2.76) | 73.35% | 73.22% (-0.13) | 6.48% | 6.05% (-0.43) |
> | STE | mnist (shadow) | 1.18 | 3.25 (+2.07) | 97.55% | 97.24% (-0.31) | 9.86% | 9.46%  (-0.40) |
>
> We observe:
>
> - **Comparable downstream behavior:** ACC and ER are highly consistent between DDPM and flow-based models.
> - **FID differences reflect modeling differences:** Flow models achieve better FID on CIFAR-10 but slightly worse on MNIST.
> - **Efficiency advantage:** Flow-based models reach similar performance with less time due to fewer timesteps (100 timesteps for training).
>
>
> **Q2**: Relationship between generation quality and number of injected datasets
>
> We thank the reviewer for pointing this out and will improve the clarity of Table 1 in the revision.
>
> We would like to clarify that **generation quality does not consistently degrade** when incorporating multiple datasets. Instead, Table 1 shows a **non-monotonic behavior** across datasets and metrics.
>
> Specifically:
> - The first row corresponds to **single-dataset baselines** (three separately trained models).
> - The second and third rows correspond to **two-dataset settings**, where CIFAR-10 is the explicit dataset, and one shadow dataset is added.
> - The fourth row ([1,1,1]) corresponds to **three datasets jointly incorporated**.
>
> In the [1,1,1] setting:
> - **CIFAR-10 achieves its best FID and highest ACC**
> - **Fashion-MNIST achieves its best ACC and lowest ER**
>
> This demonstrates that multi-dataset training does not universally harm generation quality and can even improve performance for certain datasets.
>
> At the same time, we observe degradation in some metrics (e.g., a slightly higher ER on CIFAR-10 and suboptimal performance on MNIST). This reflects an inherent trade-off introduced by increased distribution complexity. When more datasets are incorporated, the model must learn multiple heterogeneous distributions, which increases the difficulty of accurately modeling each individual distribution.
>
> This interpretation is further supported by our ablation results (**see Table R4 in the rebuttal to reviewer sZJu**), where increasing the number of shadow datasets leads to higher ER and lower ACC, consistent with increased modeling complexity.
>
> Overall, the observed behavior is not a simple degradation, but rather a trade-off between generalization and dataset-specific fidelity in multi-distribution learning.

---

> > ### Author Rebuttal · Reviewer_tHbE · 2026-04-01
> >
> > I appreciate your responses and have no further questions at this time.

---

> > > ### Author Response · Authors · 2026-04-01
> > >
> > > Dear reviewer tHbE,
> > >
> > > We sincerely thank you for the positive evaluation, highlighting multiple strengths and only a limited weakness. We are glad that the concerns have been fully addressed in the rebuttal.
> > >
> > > You selected the reply: "(a) Fully resolved - My concerns have been adequately addressed. If you select this option, please consider adjusting your score accordingly."
> > >
> > > Given this overall assessment, **we kindly invite the reviewer to consider improving the original rating on our paper if appropriate, to reflect the evaluation and disseminate our original work at the conference.**
> > >
> > > Thanks again.

---

### Official Review · Reviewer_3EaN · 2026-03-10

**Soundness:** 3
**Presentation:** 3
**Significance:** 2
**Originality:** 3
**Overall Recommendation:** 4
**Confidence:** 2

**Summary:**

This paper investigate how to inject additional information into the generation process using time-step embeddings in diffusion models as channels.
The authors observe potential unused capacity in the time-step embedding space of diffusion models and propose a Shadow Time-Step Embedding (STE) method, which introduces additional "shadow time steps" to encode auxiliary information.
This method allows diffusion models to support various behaviors based on the time-step range. Experiments show that this method can be applied to areas such as hidden information injection, dataset binding, and watermark verification.

**Compliance With Llm Reviewing Policy:**

Affirmed.

**Final Justification:**

The authors’ rebuttal adequately addressed my main concerns and clarified key points, resolving my previous doubts, and I am now inclined toward acceptance.

**Key Questions For Authors:**

1. How sensitive is this method to the choice of the shadow time step range or the number of shadow embeddings?

2. Did the authors expect this method to achieve similar results in diffusion models trained with continuous time step sampling?

3. Can this method be scaled to larger diffusion architectures (e.g., DiT-based models) or more complex datasets?

**Limitations:**

Yes

**Strengths And Weaknesses:**

Strengths:
1. This paper investigates an interesting and relatively underexplored aspect of diffusion models: the role of time-step embedding as a potential information channel.
2. The proposed shadow time-step embedding mechanism is simple and easy to implement, and can be integrated into existing diffusion architectures.

Weekness:
1. The diffusion models used for evaluation are relatively outdated. Most experiments were conducted on DDPM-type models and small datasets such as CIFAR-10 and MNIST. It would be more meaningful to evaluate the method on more modern diffusion architectures or larger datasets.
2. Evaluation of continuous-time-step diffusion models is lacking. Experiments primarily used models trained with SD1.5-style discrete-time-step sampling, where the time step is randomly selected from a fixed set of indices during training (e.g., 809 selected from a range of 1-1000). In contrast, many state-of-the-art diffusion models employ continuous-time-step sampling (e.g., t∼U(0,1)). Since the proposed method directly modifies the time-step embedding, it would be meaningful to evaluate whether the method remains effective under these continuous-time training frameworks.

---

> ### Author Rebuttal · Authors · 2026-03-31
>
> ## Response to Reviewer 3EaN.
>
> We thank the reviewer for the insightful comments, especially regarding continuous-time diffusion models and scalability to modern architectures. We address the questions below.
>
> **Q1**: Sensitivity to shadow timestep range and number of embeddings
>
> The same question is asked by other reviewers. Please kindly refer to the corresponding reply for more details.
> Here we summarize the reply to the reviewer: As discussed in the response to **Reviewer sZJu, Q5**, we have added additional ablation studies in **Table R3 and Table R4**:
>
> - **Offset size ($f_n$):**
>   We vary $f_n$ from 100 to 5000. When $f_n=100$, STE fails due to overlap in the timestep embedding space, leading to the **data overlap issue** discussed in Section 4.5.
>   When $f_n \geq 500$, all metrics (FID, ACC, ER) become stable with low variance, indicating that STE is robust once the offset exceeds a threshold.
>
> - **Number of shadow intervals:**
>   Increasing the number of embedded datasets leads to higher ER and lower ACC, reflecting the increased complexity of modeling multiple distributions. This suggests a natural trade-off between capacity and interference.
>
> Overall, STE is not highly sensitive once the offset is sufficiently large (>500), but the number of shadow distributions introduces a controllable capacity–performance trade-off.
>
>
> **Q2**: Applicability to continuous-time diffusion models
>
> We agree that evaluating STE under continuous-time frameworks is important.
>
> To partially address this, we extend STE to **flow matching models**, which are inherently **continuous-time generative models**. As shown in **Table R2 in the rebuttal to reviewer KayD**, STE achieves comparable performance (FID, ACC, ER) between DDPM (discrete-time) and flow-based models.
>
> From a methodological perspective, STE operates on the **timestep embedding function $\Phi(t)$**, rather than on the discretization itself. Therefore:
>
> - In discrete-time models: $t \in \{1, ..., T\}$
> - In continuous-time models: $t \sim U(0,1)$
>
> STE can be generalized by applying a **continuous offset transformation** (e.g., $t \rightarrow t + \delta$), which maps samples into a shifted embedding region. As long as the embedding function remains smooth (e.g., sinusoidal or learned positional encoding), the **low mutual coherence property between separated intervals still holds**, preserving distribution separability.
>
> Our flow-matching results provide empirical evidence supporting this intuition. We will further explore fully continuous-time training (e.g., VP-SDE-based models) in future work.
>
> **Q3**: Scalability to larger architectures and datasets
>
> Thank you for your suggestion. As some reviewers raised a similar question, we briefly explain it here.
>
> We have conducted additional experiments on a large-scale diffusion transformer:
>
> - **Model:** DiT-XL/2
> - **Dataset:** ImageNet (explicit) + CelebA (shadow), both at 256×256
> - **Setting:** fine-tuning with STE
>
> As shown in **Table R1 in rebuttal to reviewer KayD**, STE maintains low ER while preserving generation quality on the explicit dataset, demonstrating that the method remains effective in **large-scale architectures and datasets**.
> Also, in the appendix of our main paper, the evaluation of Stable Diffusion 1.5 results is presented.
> These results indicate that STE is **not limited to small DDPM models**, but is compatible with modern diffusion backbones.
>
> Overall, these additional experiments and analyses suggest that STE is a general mechanism rooted in the timestep embedding structure and can extend across model scales and generative paradigms.

---

> > ### Author Rebuttal · Reviewer_3EaN · 2026-04-02
> >
> > Thank you for the clarification and additional explanation. I appreciate the authors’ effort in addressing these concerns. I will take the rebuttal into account in my final assessment.

---

> > > ### Author Response · Authors · 2026-04-03
> > >
> > > Dear Reviewer 3EaN,
> > >
> > > We sincerely thank you for the positive evaluation. We are glad that the concerns have been fully addressed in the rebuttal.
> > >
> > > You selected the reply: "(a) Fully resolved - My concerns have been adequately addressed. If you select this option, please consider adjusting your score accordingly."
> > >
> > > Given this assessment regarding our rebuttal, **we kindly invite the reviewer to consider improving the original rating on our paper if appropriate, to reflect the evaluation and help disseminate our original work at the conference.**
> > >
> > > Thanks again.

---

### Official Review · Reviewer_sZJu · 2026-03-10

**Soundness:** 2
**Presentation:** 3
**Significance:** 3
**Originality:** 3
**Overall Recommendation:** 4
**Confidence:** 3

**Summary:**

This paper introduces Shadow Timestep Embedding (STE), a novel mechanism for information injection in diffusion models by exploiting the underutilized temporal embedding space. By incorporating shadow offsets into the original timestep domain, STE enables the encoding and generation of independent data distributions. Experiments demonstrate that STE preserves generation fidelity while facilitating isolated distribution learning and achieving high success rates in security applications (information injection attack and watermark verification).

**Compliance With Llm Reviewing Policy:**

Affirmed.

**Key Questions For Authors:**

See the weaknesses.

**Limitations:**

yes

**Strengths And Weaknesses:**

Strengths:
1. The paper focuses on the intrinsic properties of the timestep embedding in diffusion models, an underexplored aspect in previous research with significant implications for model security risks.
2. The core idea of mapping different timestep embeddings to multiple data distributions is interesting, and the proposed method enables effective dual-purpose applications in attacks and defenses.
3. The paper is written in a clear structure and easy to follow.

Weaknesses:
1. Current experiments are confined to small-scale models and low-resolution datasets such as CIFAR-10 and MNIST. More extensive evaluations on larger datasets and models are necessary to validate the method's scalability and effectiveness. Moreover, since the method requires model retraining, its practical deployment feasibility for larger diffusion models is limited.
2. The paper lacks investigation into the method's generalization capability across different diffusion model architectures. For more advanced structures such as MM-DiT, it remains unclear whether timestep embeddings exhibit similar properties and whether the proposed method maintains its effectiveness.
3. The method exhibits worse robustness (especially to crop/brightness) than other watermarking methods (Table 3).
4. Baselines are not very comprehensive. More diffusion-based steganography baselines should be included to better validate its performance in embedding and recovering hidden information.
5. The paper lacks some key ablation studies, including the size of offset $f_n$ and the number of shadow intervals corresponding to different datasets.
6. Several evaluation metrics are insufficiently defined. For example, the paper fails to specify the classifier for Exposure Rate calculations and detector for Watermark Detection Accuracy.

---

> ### Author Rebuttal · Authors · 2026-03-31
>
> ## Response to Reviewer sZJu.
>
> Thank you for the constructive feedback. We are encouraged that the reviewer recognizes the novelty of exploring timestep embeddings and their implications for model security.
>
> **Q1 & Q2**: Scalability and Generalization to Different Architectures
>
> To address scalability and generalization, we have conducted additional experiments and report the results in Tables R1 and R2 in the rebuttal to **reviewer KayD**: (1) *Large-scale model (DiT):* We evaluate STE on a pre-trained **DiT-XL/2** model using ImageNet (256×256) as the explicit dataset in Table R1. (2) *Alternative generative model:* We extend STE from DDPM to **flow-based diffusion models** and report results in Table R2.
>
> These results show that STE maintains a low Exposure Rate and preserves generation quality in large-scale settings, and that it achieves performance comparable to DDPM and flow-based models, indicating that the method is not tied to a specific diffusion formulation.
>
> **Q3**: Watermark Robustness
>
> Our STE is less robust to cropping for a reason: STE's current implementation uses an *explicit watermark* (a pattern placed in the top-right corner). If the cropped region removes this area, detection accuracy naturally drops. This limitation stems from the *choice of watermark representation*, not from the STE mechanism itself. In future work, we plan to incorporate *implicit watermarking strategies* (e.g., in the frequency domain) to improve robustness.
> As for *Brightness* robustness, the detection accuracy drops by only 0.01 compared with Tree-Ring, which is slightly worse than the baseline.
>
> **Q4**: Baselines
>
> Existing diffusion-based steganography methods [1,2] mainly focus on the *image-level*, where information is embedded in the generated outputs. In contrast, our method operates at the *model level*, leveraging the representational capacity of timestep embeddings to encode separate data distributions. This leads to a trade-off: (1) *Advantages:* our method has a stronger capacity to encode complex information (entire distributions rather than pixel-level signals) and more flexible control via timestep selection. (2) *Limitations:* Lower efficiency compared to training-free or image-level methods. Therefore, STE explores a complementary and previously underexplored direction in diffusion-based steganography.
>
> **Q5**: Ablation Studies
>
> We have added additional ablation studies on both *offset size $f_n$* and *number of shadow intervals*.
>
> *Effect of offset size $f_n$, table R3*: We use CIFAR-10/MNIST as the explicit/shadow dataset, and vary $f_n$ from 100 to 5000. When $f_n \geq 500$, the performance becomes stable across all metrics. We also report the mean ± std (excluding $f_n=100$), showing that STE is robust once the offset exceeds a threshold.
>
> *Effect of number of shadow intervals, table R4*: CIFAR-10 is used as the explicit dataset, and shadow datasets are constructed from different ImageNet subsets (each containing 10 classes). Results in table R4 show that FID is non-monotonic with increasing interval. ACC decreases, and ER increases as more datasets are embedded. This reflects the increased complexity of modeling multiple distributions simultaneously.
>
> *Table R3. The ablation experiments on different time offsets $f_n$.*
> | $f_n$ | FID (Explicit) ↓ | FID (Shadow) ↓ | ACC (Explicit) ↑ | ACC (Shadow) ↑ | ER (Explicit) ↓ | ER (Shadow) ↓ |
> |--|--|--|--|--|--|--|
> | 100   | 25.33 | 145.34 | 69.28% | 42.27% | 12.15% | 18.61% |
> | 500   | 19.91 | 1.30   | 74.56% | 97.44% | 6.50%  | 10.00% |
> | 1000  | 22.20 | 1.18   | 73.35% | 97.55% | 6.48%  | 9.86%  |
> | 1500  | 21.46 | 1.33   | 75.55% | 97.41% | 6.57%  | 9.95%  |
> | 2000  | 19.66 | 1.56   | 76.63% | 97.64% | 6.70%  | 9.97%  |
> | 5000  | 20.24 | 1.34   | 76.15% | 97.49% | 6.42% | 9.78% |
> | *Mean ± Std* | *20.69 ± 0.97* | *1.34 ± 0.12* | *75.25% ± 1.17* | *97.51% ± 0.08* | *6.53% ± 0.10* | *9.91% ± 0.08* |
>
> *Table R4. Impact of different numbers of intervals.*
> | #Intervals | FID ↓  | ACC ↑  | ER ↓  |
> |--|--|--|--|
> | 0 | 24.38  | 73.41% | 8.92% |
> | 1 | **19.37**  | **74.77%** | **8.52%** |
> | 2 | 21.53  | 71.44% | 9.85% |
> | 4 | 20.01  | 70.71% | 11.15%|
> | 8 | 23.90  | 68.34% | 13.58%|
>
> **Q6**: Metrics Clarification
>
> *Accuracy:* A classifier trained on the corresponding dataset is used to evaluate generated samples.
>
> *Exposure Rate:* ER measures the probability that samples generated from the explicit timestep are classified as shadow datasets.
>
> *Watermark Detector:* A classifier trained with watermarked images (with shield pattern) and clean images.
>
> [1] Yu, J., Zhang, X., Xu, Y., & Zhang, J. (2023). Cross: Diffusion model makes controllable, robust and secure image steganography. Advances in Neural Information Processing Systems, 36, 80730-80743.
> [2] Xu, Y., Zhang, X., Meng, X., Mou, C., & Zhang, J. (2025, June). Diffusion-based hierarchical image steganography. In 2025 IEEE International Conference on Multimedia and Expo (ICME) (pp. 1-6). IEEE.

---

> > ### Author Rebuttal · Reviewer_sZJu · 2026-04-04
> >
> > Thank you for the rebuttal. After reviewing your responses, I have no remaining questions and will keep my original score.

---

> > > ### Author Response · Authors · 2026-04-05
> > >
> > > Dear reviewer,
> > >
> > > We sincerely thank you for the positive evaluation. We are glad that the concerns have been fully addressed in the rebuttal.
> > >
> > > You selected the reply: "(a) Fully resolved - My concerns have been adequately addressed. If you select this option, please consider adjusting your score accordingly."
> > >
> > > Given this overall assessment, we __kindly invite the reviewer to consider improving the rating on our paper if appropriate, to reflect the evaluation and disseminate our original work at the conference.__
> > >
> > > Thanks again.

---

### Official Review · Reviewer_KayD · 2026-03-11

**Soundness:** 3
**Presentation:** 3
**Significance:** 3
**Originality:** 3
**Overall Recommendation:** 5
**Confidence:** 3

**Summary:**

This paper introduces a novel method for injecting information into models through shadow timesteps. The authors demonstrate that the standard timestep interval [0, T] can be extended to [f_n, T+f_n], where each of the n intervals can be trained on its own individual data distribution P_n. During generation, when using interval k with the same sampler, the model can generate data from the corresponding distribution P_k.

The authors provide both theoretical and experimental evidence that this training approach prevents distribution leakage between intervals, and that the timestep embedding space enables clear separation between different distributions. This method has potential applications in both covert learning of undesirable information and copyright verification through watermarked image generation.

**Compliance With Llm Reviewing Policy:**

Affirmed.

**Final Justification:**

The authors have addressed my primary questions regarding the scalability and application of the flow-matching model. They demonstrated that, in both scenarios, the STE can effectively separate different distributions while maintaining high generalization. In my opinion, the paper highlights a very interesting property of the model: the timestep mechanism is sufficiently robust to train multiple distributions across different timestep intervals while fine-tuning the same diffusion weight. Therefore, I recommend accepting this paper and am raising my score from 4 to 5.

It would be very interesting to investigate whether this mechanism is applicable to large and complex models such as FLUX or SD-XL. However, I believe that conducting such an experiment could be highly computationally intensive and may fall outside the scope of this scientific investigation.

**Key Questions For Authors:**

Please provide a detailed response addressing the weaknesses outlined above, particularly regarding scalability to larger datasets, applicability to flow-based models, and performance in fine-tuning scenarios.

**Limitations:**

yes

**Strengths And Weaknesses:**

Strengths:
1) The paper is clearly written and easy to follow
2) The paper explores a novel approach to information injection through timestep embeddings, addressing a gap in the current literature
3) The experiments demonstrate that the method effectively enables learning distinct distributions across different timestep intervals without significant leakage between distributions

Weaknesses:
1) The paper evaluates the method exclusively on small-scale datasets, raising concerns about potential data memorization. It remains unclear whether the method's performance - particularly the absence of leakage between different data distributions - can be maintained when scaled to larger datasets. Also, I did not find information about model size, which is also can be very important for the method performance.
2) The paper focuses solely on DDPM-based diffusion models and demonstrates that the method is quite sensitive to sampler choice. However, since current state-of-the-art models are flow-based, it would be valuable to investigate how the method performs in these case.
3) As far as I understand, the method is evaluated only during the pre-training stage. It would be particularly interesting to examine its performance in fine-tuning scenarios - for instance, taking a pre-trained model (such as SD-XL or FLUX) and fine-tuning it on shadow timestep intervals.

---

> ### Author Rebuttal · Authors · 2026-03-31
>
> ## Response to Reviewer KayD.
> We sincerely thank the reviewer for the constructive feedback. We are encouraged by the reviewer's finding that the method is novel, clearly presented, and potentially impactful. Below, we address the key concerns raised by the reviewer.
>
> **Q1 & Q3**: Scalability to Larger Models and Fine-tuning Scenarios
>
> **Scalability to large-scale datasets/models and fine-tuning on pre-trained models.**
> To evaluate scalability and applicability to modern architectures, we conduct experiments on a large-scale pre-trained diffusion transformer, **DiT-XL/2** [1], trained on ImageNet at 256×256 resolution. STE is implemented via fine-tuning, with ImageNet as the explicit dataset and CelebA (256×256) as the shadow dataset. We generate 30K samples and evaluate both generation quality (FID) and separation (ER) using a classifier trained on ImageNet and CelebA.
>
> Table R1 demonstrates that STE can effectively encode and separate multiple data distributions even in large-scale pre-trained models, while maintaining strong generation fidelity. This suggests that STE is not limited to small-scale DDPM settings but generalizes well to modern large-scale generative backbones.
>
> *Table R1. The performance of STE on DiT.*
> | Setting | FID (Imagenet)↓ | FID (CelebA)↓ | ER (Imagenet) ↓| ER (CelebA)↓ |
> |:---------:|:----------------:|:--------------:|:---------------:|:-------------:|
> | Clean   | 2.27           | 9.64       |   0.10%  |0.10%  |
> | STE     | 3.22           | 3.30         |   0.13%  | 0.15% |
>
> STE introduces only a moderate increase in FID on ImageNet (from 2.27 to 3.22). The change in CelebA FID can be attributed to differences in training settings: in the clean setting, CelebA is trained from scratch for 50K steps, whereas in the STE setting, the model is fine-tuned on a pre-trained model. So, STE fine-tuning typically achieves better generation quality under the same number of training steps.
>
> Meanwhile, ER remains extremely low, indicating minimal leakage between distributions. Notably, the ER values differ significantly between Table R1 and Table 1 (in the paper) due to the number of classes in the evaluation setting. In Table R1, the DiT model is trained on ImageNet-1K, and the classifier also operates over 1,000 classes, leading to substantially lower ER values. In Table 1, the datasets contain only 10 classes, and ER fluctuates around 10%, which corresponds to the expected accuracy under random or ambiguous predictions. This difference is therefore primarily due to the scale of the classification task rather than the behavior of STE.
>
> Moreover, we already conducted a preliminary fine-tuning experiment on Stable Diffusion v1.5, as shown in Figure B4 of the paper's appendix. It demonstrates that explicit timesteps still produce normal outputs, while shadow timesteps activate the injected pattern.
>
> **Model size clarification.**
> For completeness, we clarify the model configurations:
>
> **DDPM experiments:** U-Net backbone with ~37M parameters, 4 resolution levels (128, 256, 256, 256), 2 residual blocks per level, with attention at lower resolutions.
>
> **Flow-based experiments:** same U-Net backbone as DDPM trained with Flow Matching [2].
>
> **Large-scale experiments:** DiT-XL/2 architecture.
>
> These results collectively demonstrate that STE generalizes to small and large-scale settings and remains effective under both scaling and fine-tuning.
>
> **Q2**: Applicability to Flow-based Models
>
> In the rebuttal, we extend STE to flow-based generative models using the flow matching framework [2]. Table R2 summarizes the comparison between DDPM and flow-based models:
>
> *Table R2. The performance of STE on DDPM and Flow Matching.*
> | Setting | Dataset | FID (DDPM) ↓| FID (Flow) ↓| ACC (DDPM)↑ | ACC (Flow) ↑ | ER (DDPM)↓ | ER (Flow) ↓|
> |------|--------|:----------:|:----------:|:----------:|:----------:|:---------:|:---------:|
> | Clean | cifar-10 (single) | 24.38 | 20.45 (-3.93) | 73.41% | 73.56% (+0.15) | 5.39% | 6.43% (+1.04) |
> | Clean | mnist (single) | 1.59 | 3.22 (+1.63) | 98.33% | 98.80% (+0.47) | 9.71% | 8.73% (-0.98) |
> | STE | cifar-10 (Explicit) | 22.20 | 19.44 (-2.76) | 73.35% | 73.22% (-0.13) | 6.48% | 6.05% (-0.43) |
> | STE | mnist (shadow) | 1.18 | 3.25 (+2.07) | 97.55% | 97.24% (-0.31) | 9.86% | 9.46%  (-0.40) |
>
> We observe that (1) ACC and ER are highly consistent between DDPM and flow-based models. (2) Flow models achieve better FID on CIFAR-10 but slightly worse on MNIST. (3) Flow-based models reach similar performance with less time due to fewer timesteps (100 timesteps for training). The result means STE transfers naturally to flow-based models.
> Importantly, the effectiveness of STE relies on the **embedding separability in timestep space**, which is independent of the specific generative formulation.
>
> [1] Peebles, W., & Xie, S. (2023). Scalable diffusion models with transformers. ICCV.
> [2] Lipman, Y., et al. (2022). Flow matching for generative modeling. arXiv:2210.02747.

---

> > ### Author Rebuttal · Reviewer_KayD · 2026-04-04
> >
> > Thank you for your clarifications and the additional experiments. I have no further questions. I’m raising my score from 4 to 5

---

> > > ### Author Response · Authors · 2026-04-05
> > >
> > > Dear reviewer,
> > >
> > > We sincerely thank you for the positive evaluation. We are glad that the concerns have been fully addressed in the rebuttal.
> > >
> > > Thanks again.

---

### Decision · Program_Chairs · 2026-04-30

**Decision:**

Accept (regular)

**Comment:**

This paper investigates the utility of using time-step embeddings to encode auxilliary information in diffusion models. To this end, they introduce shadow timestep embeddings, allowing the diffusion models to "multiplex" between various behaviors indicated by these embeddings. They show that this method can be used for hidden information injection, dataset binding and watermark verification.


All the reviewers are positive about this paper. The main concerns about pre-training only small scale DDPMs was addressed by ablations studies where the authors fine-tuned large scale flow models to show similar behavior. Based on the reviews, this appears to be a significant contribution to the ML literature. I recommend acceptance.